# *Thymus vulgaris* Essential Oil and Its Biological Activity

**DOI:** 10.3390/plants10091959

**Published:** 2021-09-19

**Authors:** Lucia Galovičová, Petra Borotová, Veronika Valková, Nenad L. Vukovic, Milena Vukic, Jana Štefániková, Hana Ďúranová, Przemysław Łukasz Kowalczewski, Natália Čmiková, Miroslava Kačániová

**Affiliations:** 1Institute of Horticulture, Faculty of Horticulture and Landscape Engineering, Slovak University of Agriculture, Tr. A. Hlinku 2, 94976 Nitra, Slovakia; veronika.valkova@uniag.sk (V.V.); xcmikova@uniag.sk (N.Č.); 2Institute of Applied Biology, Faculty of Biotechnology and Food Sciences, Slovak University of Agriculture, Tr. A. Hlinku 2, 94976 Nitra, Slovakia; petra.borotova@uniag.sk; 3AgroBioTech Research Centre, Slovak University of Agriculture, Tr. A. Hlinku 2, 94976 Nitra, Slovakia; jana.stefanikova@uniag.sk (J.Š.); hana.duranova@uniag.sk (H.Ď.); 4Department of Chemistry, Faculty of Science, University of Kragujevac, 34000 Kragujevac, Serbia; nvchem@yahoo.com (N.L.V.); milena.vukic@pmf.kg.ac.rs (M.V.); 5Department of Food Technology of Plant Origin, Poznań University of Life Sciences, 31 Wojska Polskiego St., 60-624 Poznań, Poland; przemyslaw.kowalczewski@up.poznan.pl; 6Department of Bioenergy, Food Technology and Microbiology, Institute of Food Technology and Nutrition, University of Rzeszow, 4 Zelwerowicza St, 35601 Rzeszow, Poland

**Keywords:** *Thymus vulgaris*, biofilm, DPPH, *P. fluorescens*, *S. enteritidis*

## Abstract

*Thymus vulgaris* essential oil has potential good biological activity. The aim of the research was to evaluate the biological activity of the *T. vulgaris* essential oil from the Slovak company. The main components of *T. vulgaris* essential oil were thymol (48.1%), *p*-cymene (11.7%), 1,8-cineole (6.7), γ-terpinene (6.1%), and carvacrol (5.5%). The antioxidant activity was 85.2 ± 0.2%, which corresponds to 479.34 ± 1.1 TEAC. The antimicrobial activity was moderate or very strong with inhibition zones from 9.89 to 22.44 mm. The lowest values of MIC were determined against *B. subtilis*, *E. faecalis,* and *S. aureus*. In situ antifungal analysis on bread shows that the vapor phase of *T. vulgaris* essential oil can inhibit the growth of the microscopic filamentous fungi of the genus *Penicillium*. The antimicrobial activity against *S. marcescens* showed 46.78–87.80% inhibition at concentrations 62.5–500 µL/mL. The MALDI TOF MS analyses suggest changes in the protein profile of biofilm forming bacteria *P. fluorescens* and *S. enteritidis* after the fifth and the ninth day, respectively. Due to the properties of the *T. vulgaris* essential oil, it can be used in the food industry as a natural supplement to extend the shelf life of the foods.

## 1. Introduction

*Thymus vulgaris* is a flowering plant of the Lamiaceae family. The origin of this plant is in southern Europe. It is an evergreen shrub with small, strongly aromatic, gray-green leaves and with purple or pink bunches of flowers [1]. Thyme was used in folk medicine for centuries for its significant antimicrobial and anti-inflammatory effects. It is often used in cooking as a seasoning [2]. *T. vulgaris* is characterized by chemical polymorphism according to the main volatile component. There are six known chemotypes of *T. vulgaris* essential oils: geraniol, linalool, α-terpineol, tujanol-4, thymol, and carvacrol. In the vast majority of plants and subsequent essential oils, one chemotype occurs, but in some, we can find two to three different chemotypes [3,4]. Due to the antimicrobial features of the main constituents, *T. vulgaris* essential oils are effective not only against bacteria and yeasts but also show inhibition activity against microscopic filamentous fungi [5,6].

Bacteria are very rarely found in planktonic lifeform as they are often exposed to an adverse environment. For higher resistance, bacteria begin to form a biofilm which is a 3D structure surrounded by an extracellular matrix of polysaccharides [7]. These bacterial forms are difficult to control and are highly resistant to antibiotics [8].

*Salmonella enteritidis* is a gram-negative bacterium (G^−^) that causes gastrointestinal problems due to contamination of predominantly animal products such as meat and eggs [9]. Its cells can form biofilms and adhere to various surfaces that are in contact with food. The majority of the biofilms occur in food processing industries, on work surfaces such as stainless steel, glass, rubber, and polyurethane [10]. As a result, cross-contamination of products may occur. Due to biofilm formation, the bacteria are more resistant to commonly used disinfectants which increase the risk of the contamination [11].

*Pseudomonas fluorescens* is a gram-negative foodborne pathogen that causes the spoilage of foods with a high-water content. Compared to the other pathogens, it can grow even at low temperatures [12]. In the dairy industry, it is a source of contamination for milk and dairy products. *P. fluorescens* is a suitable and recognized model organism for the study of biofilms [13].

The most commonly used techniques for studying biofilms are microscopy-based methods, which are very limited. Mass spectrometry is a suitable option for the study of biofilms [14]. The MALDI-TOF MS Biotyper is a suitable and new method for the analysis of phenotypic differences in biofilm progression. Additionally, the changes in the properties of the growth related to the different surfaces was detectable [15]. To this date, only a few authors have followed the development and structure of biofilms using MALDI-TOF MS Biotyper [15,16,17].

Essential oils from medicinal plants were used since ancient times to treat certain diseases due to their antimicrobial effects. Currently, there is a trend of increasing antibiotic resistance of many microorganisms, and it is necessary to look for alternative compounds to eliminate them. Due to their antimicrobial effects, essential oils represent a promising potential as a natural alternative to antibiotics [18].

This study’s aim was to analyze the biological activity of the essential oil from *T. vulgaris*. To determine the antioxidant, antimicrobial, antibiofilm activity, as well as the chemical composition, of the essential oil. Moreover, the evaluation of the changes in biofilm structure on glass and wood surfaces on microorganisms *Salmonella enteritidis* and *Pseudomonas fluorescens* using MALDI-TOF MS Biotyper was performed. We also focused on the antifungal effect of the vapor phase of essential oil on a food model.

## 2. Results

### 2.1. Chemical Composition of T. vulgaris Essential Oil

Using gas chromatography/mass spectrometry (GC/MS) and gas chromatography (GC-FID), we detected the main components of *T. vulgaris* essential oil is thymol 48.1%, *p*-cymene 11.7%, 1,8-cineole 6.7%, *γ*-terpinene 6.1%, and carvacrol 5.5% (Table 1). Based on the chemical composition, we classify the essential oil *T. vulgaris* into the thymol chemotype. 

### 2.2. Antioxidant, Antimicrobial Activity, and Minimal Inhibition Concentration (MIC)

The antioxidant activity of *T. vulgaris* essential oil was determined to be 85.2 ± 0.2% using the DPPH radical and expressed as the equivalent of the standard substance Trolox 479.34 ± 1.1 TEAC. The antimicrobial activity was evaluated by the disk diffusion test evaluating the effect against gram-positive bacteria *(Bacillus subtilis* CCM 1999, *Enterococcus faecalis* CCM 4224, *Staphylococcus aureus,*), gram-negative bacteria (*Pseudomonas aeruginosa* CCM 3955, *Yersinia enterocolitica* CCM 7204, *Salmonella enterica* subsp. *enterica* ser. Enteritidis CCM 4420, *Serratia marcescens* CCM 8588), yeast (*Candida krusei* CCM 8271, *Candida albicans* CCM 8261, *Candida tropicalis* CCM 8223, *Candida glabrata* CCM 8270), and biofilm-forming bacteria (*Salmonella enteritidis*, *Pseudomonas fluorescens*). In all tested microorganisms, including biofilm-forming bacteria, we observed moderate and very strong inhibitory activity except in *C. tropicalis*, which showed moderate inhibitory activity slightly below the threshold value of strong activity. MIC 50 and MIC 90 were determined by analysis of the minimum inhibitory concentrations. Low values of MIC 50 (12.12–16.56 µL/mL) and MIC 90 (16.43–19.26 µL/mL) were found in *B. subtilis, E. faecalis,* and *S. aureus*. The highest MIC 50 and MIC 90 values were determined for *S. enteritidis* biofilm. Moderate MIC 50 and MIC 90 values were determined for *S. enterica* subs. *enterica* ser. Enteritidis, *P. aeruginosa, Y. enterocolitica, C. albicans, C. krusei, C. tropicalis, C. glabrata, S. marcescens,* and *P. fluorescens* biofilm. Details of the results of antimicrobial activity and minimum inhibitory concentrations are given in Table 2.

### 2.3. Analysis of Biofilm Developmental Phases and Evaluation of Molecular Differences on Different Surfaces Using MALDI-TOF MS Biotyper

The effect of *T. vulgaris* essential oil on the molecular structure and growth inhibition of *S. enteritidis* and *P. fluorescens* biofilms was evaluated using MALDI TOF MS Biotyper. The spectra of biofilms and planktonic cells in the control group developed identically and, therefore, the spectra of planktonic cells were used for greater clarity than the control spectrum. For each day, two experimental spectra from different surfaces (glass, wood) and a planktonic spectrum representing the development of the control group are shown.

Figure 1 shows the mass spectra of *S. enteritidis* biofilm during the individual days of the experimental evaluation.

The mass spectra obtained on the third day of the experiment represent very small differences in peaks between the experimental and control planktonic spectra (Figure 1). From day 5 of the experiment, we noticed differences between the two experimental spectra and the control spectrum. During 5 and 7 days, we noticed a more significant difference in the experimental group on wood. From day 9 to day 14, the differences were significant in both experimental groups compared to the control. Due to the influence of the essential oil *T. vulgaris*, we were able to observe changes in the protein spectrum of the biofilm. This finding suggests that the essential oil disrupts biofilm homeostasis leading to the degradation of this form of microorganism.

To visualize the similarity of mass spectra, a dendrogram was constructed based on MSP distances (Figure 2). The control groups and spectra of young biofilms (3 days) had the shortest distance together with planktonic cells. From day 5, we could see an increase in the distance of MSP experimental groups. During days 5 and 7, it can be seen that the experimental group on glass has a much shorter MSP distance than the experimental group on wood. This trend was maintained with a smaller difference throughout the duration of the experiment. The distance MSP of control groups from all tested days was significantly shorter than in the experimental groups. The increasing length of the MSP in experimental groups suggests changes in the protein profile of the bacterial biofilm of *S. enteritidis*.

Figure 3 shows the mass spectra of *P. fluorescens* biofilm during the individual days of the experimental evaluation.

In the first days of the experiment, the experimental groups of *P. fluorescens* biofilm with the addition of *T. vulgaris* essential oil had similar mass spectra as the control group (Figure 3). From the third to the seventh day of the experiment, there was no change in the protein spectrum of the experimental group due to the action of *T. vulgaris* essential oil. From day 7 of the experiment, we recorded lower peaks of the experimental group compared to the control group, which indicates the influence of the essential oil in the experimental group. From the ninth day, the appearance of the spectra of the experimental group changed more significantly. Due to the change in the protein spectrum of the experimental group compared to the control spectrum, we evaluate the effect of the essential oil as positive for disrupting the viability of the biofilm structure of *P. fluorescens* during the longer exposure.

To visualize the similarity of mass spectra, a dendrogram was constructed based on MSP distances. The control groups and spectra of biofilms of the third, fifth, and seventh day had the shortest distance together with planktonic cells (Figure 4). From day 9, we observed an increase in the distance of MSP experimental groups. During days 12 and 14, the MSP distance of the experimental group was the longest. These findings are evidence of the effect of *T. vulgaris* essential oil on altering the molecular structure of biofilms. Based on the constructed dendrogram, it can be stated that the MSP distance of the control groups was significantly shorter than the distance of the experimental groups from day 9, which confirms the inhibitory effect on the development of *P. fluorescens* biofilm at a longer exposure.

### 2.4. Antimicrobial Analysis of Bread In Situ

In situ antimicrobial analysis on bread shows that microscopic filamentous fungi of the genus *Penicillium* were inhibited by all tested concentrations (Table 3). The highest percentage of inhibition at 62.5 µL/L was 82.17% for *P. commune*. The lowest rate of inhibition at this concentration was observed for *P. chrysogenum*. At the concentration of 125 μL/L, the inhibition rate was 80–89% for all microorganisms. At the concentration of 250 μL/L, the inhibition was 92–96%. At the highest tested concentration, the inhibition was 98–100%. For *P. glabrum*, a significance was noted between concentration 62.5 µL/L compared to 250 µL/L and 500 µL/L. For *P. chrysogenum*, differences were visible between 62.5 µL/L and the remaining concentrations. For *P. expansum*, no significant differences were observed between the tested concentrations. For *P. commune*, a significant difference was observed between concentrations of 62.5 µL/L and 500 µL/L. These findings suggest the inhibitory effects of *T. vulgaris* against the potentially pathogenic fungi.

### 2.5. In Situ Antimicrobial Analysis on Carrots

From in situ analysis on carrots, the effect of the vapor phase of *T. serpyllum* essential oil against *S. marcescens* was recorded at all tested concentrations (Table 4). The highest inhibition was observed at a concentration of 500 µL/L by 87.8%. The lowest inhibition rate was at a concentration of 125 µL/L by 46.78%. The vapor phase of EO from *T. vulgaris* has the inhibitory effect on the growth of the bacteria on the food model.

## 3. Discussion

Plants, and subsequently essential oils of *T. vulgaris*, can have different chemotypes. Using chemical composition analysis, we verified that the essential oil we tested belongs to the thymol chemotype. Al-Asmari et al. [19] analyzed the essential oil of *T. vulgaris* according to its chemical composition was also the thymol type but as other main components they determined furan 12.19% and *p*-cymene 2.78%. Micucci et al. [20] determined essential oil as thymol-type, and as other major components carvacrol and *p*-cymene was observed. Thymol and carvacrol are terpenoids that commonly occur as major components of essential oils [21]. They are recognized as safe in food by the Food and Drug Administration [22]. Thymol and carvacrol are very effective against foodborne pathogens such as *Salmonella* spp. and *Staphylococcus aureus* that can produce biofilms by which they adhere to various surfaces and thus endanger food operations [23,24].

For the analysis of antioxidant activity of essential oil *T. vulgaris*, we used DPPH radical dissolved in methanol. The percentage inhibition of DPPH free radical for *T. vulgaris* essential oil was determined to be 85.2 ± 0.2%, which corresponds to 479.34 ± 1.1 TEAC. Bistgani et al. [25] determined the antioxidant activity of the methanol extract of *T. vulgaris* 69.7 ± 4.8%. Punya et al. [26] set the DPPH radical scavenging activity at 78.73%. Kulisic et al. [27] set the percentage inhibition at 91.30 ± 0.30%. Although the method of DPPH free radical scavenging is not standardized and each of these authors performed it with modifications, our results agree that *T. vulgaris* shows strong antioxidant activity.

Using the disk diffusion method, we found moderate to very strong antimicrobial activity in all tested microorganisms, including biofilm-forming bacteria. For *C. tropicalis*, we found a borderline moderate inhibitory activity. Al Maqtari [28] evaluated the effect of *T. vulgaris* against *B. subtilis*, *S. aureus*, *P. aeruginosa,* and *C. albicans* by disk diffusion method and recorded a zone of inhibition more than 20 mm in all tested microorganisms, which is considered to be a very strong inhibition. Borugă et al. [29] evaluated the effect against *S. aureus*, *E. faecalis*, *C. albicans*, *S. typhimurium,* and *P. aeruginosa* and recorded zones of inhibition more than 13 mm when 10 μL per disc was applied. They rated the effect of the essential oil as very strong. Rota et al. [30] recorded zones of inhibition more than 20 mm in the tested microorganisms and evaluated the essential oil of *T. vulgaris* as very effective. Boukhatem et al. [31] reported a marked inhibition of the genus *Candida* by *T. vulgaris* essential oil. Our findings on the strong inhibitory activity of *T. vulgaris* essential oil are consistent with the authors work above.

Nikolić et al. [32] evaluated the effect of *T. vulgaris* against genus *Candida* strains where they found MIC 80–160 µL/mL and MBC 160–320 µL/mL. Jafri and Ahmad [33] evaluated the effect of *T. vulgaris* essential oil on the elimination of biofilm produced by the genus *Candida*. They found that the essential oil has lower MIC values than the tested antibiotics. Al-Shuneigat et al. [34] found in their study that *T. vulgaris* essential oil showed strong antimicrobial and antibiofilm effects MIC values were 0.0625–2% *v*/*v*.

Myszka et al. [35] evaluated the effect of *T. vulgaris* against *P. fluorescens* biofilm and found that the essential oil can degrade the biofilm formed on stainless steel. Čabarkapa et al. [36] evaluated the effect of several essential oils, including *T. vulgaris*, against *S. enteritidis* biofilm. All tested essential oils showed inhibition of *S. enteritidis* biofilm formation at subminimum concentrations of essential oils. Al-Shuneigat et al. [34] evaluated the effect of *T. vulgaris* against *P. aeruginosa* biofilm and found that a very small amount is sufficient to eliminate both biofilms and planktonic cells. De Oliveira et al. [37] stated that *T. vulgaris* essential oil has the potential to be used as a biofilm control agent. MALDI-TOF MS Biotyper was used in only a small number of publications. Li et al. [12] used this method to analyze *B. subtilis* biofilm and to determine the spatial distribution of specific peptides and lipopeptides that are produced in biofilms. Kubesová et al. [38] used MALDI-TOF MS to analyze the biofilm produced by the genus *Candida*. Rams et al. [39] found that the phenotypic identification of culturable *P. gingivalis* biofilms is 100% accurate using MALDI-TOF MS because of the differences in the protein profile. The changes in the mass spectra profile were demonstrated by MALDI-TOF in Kačániová et al. [40,41] where the inhibitory effects on the biofilm of *Coriandrum sativum* and *Citrus aurantium* EOs was detected. Use of MALDI-TOF can be a fast and easy method for the assessment of the biofilm growth and degradation due to the structural and molecular changes. Kloucek et al. [42] found out that vapor phase essential oils represent a suitable alternative to antimicrobials in the food industry due to the need of lower concentrations of EO than during use of contact effect of the liquid phase. Reyes-Jurado et al. [43] evaluated the effect of the vapor phase of *T. vulgaris* essential oil against filamentous microscopic fungi. Based on the antimicrobial activity of the vapor phase, they found that the essential oil has the potential to be used to protect packaged foods. Mani López et al. [44] found that vapor phase essential oils inhibited the growth of microscopic filamentous fungi of the genus *Penicillium* on bread and it would be appropriate to verify the effect on sensory properties.

## 4. Materials and Methods

### 4.1. Essential Oil

*T. vulgaris* EO of thymol chemotype was obtained from Hanus, s.r.o. (Nitra, Slovakia). The EO was prepared by steam distillation of the partially dried stalk.

### 4.2. Chemical Characterization of T. vulgaris EO by Gas Chromatography/Mass Spectrometry (GC/MS) and Gas Chromatography (GC-FID)

GC/MS analysis of *T. vulgaris* EO was performed using the Agilent 6890N gas chromatograph (Agilent Technologies, Santa Clara, CA, USA) coupled to a quadrupole mass spectrometer 5975B (Agilent Technologies, Santa Clara, CA, USA). A HP-5MS capillary column (30 m × 0.25 mm × 0.25 µm). The temperature program was set from 60 °C to 150 °C (increasing rate 3 °C/min) and from 150 °C to 280 °C (increasing rate 5 °C/min). The total run time of the program was 60 min. Helium 5.0 was used as the carrier gas with a flow rate of 1 mL/min. The injection volume was 1 µL (EO sample was diluted in pentane), while the split/splitless injector temperature was set at 280 °C. The investigated sample was injected in the split mode with split ratio at 40.8:1. Electron-impact mass spectrometric data (EI-MS; 70 eV) were acquired in scan mode over the m/z range 35–550. MS ion source and MS quadrupole temperatures were 230 °C and 150 °C, respectively. Acquisition of data started after 3 min of solvent delay time. GC-FID analyses were performed on the Agilent 6890N gas chromatograph coupled to a FID detector. Column (HP-5MS) and chromatographic conditions were the same as for GC-MS. The temperature of the FID detector was set at 300 °C.

The individual volatile constituents of *T. vulgaris* EO sample were identified according to their retention indices [45] and they were compared with the reference spectra (Wiley and NIST databases). The retention indices were determined experimentally by a standard method that included retention times of n-alkanes (C6-C34) injected under the same chromatographic conditions [46]. The percentages of the identified compounds (amounts higher than 0.1%) were derived from their GC peak areas.

### 4.3. Determination of Antioxidant Activity

The radical scavenging of 2,2-diphenyl-1-picrylhydrazyl (DPPH, Sigma Aldrich, Germany) was used to measure the antioxidant activity of *T. vulgaris* EO. The solution of DPPH (0.025 g/L dissolved in methanol) was adjusted to absorbance 0.7 at wavelength 515 nm. 5 μL of EO sample was added to 195 μL DPPH solution in 96 well microplate and the reaction solution was incubated for 30 min in the dark with continuous shaking at 1000 rpm. The antioxidant activity was expressed as the percentage of DPPH inhibition and was calculated according to the formula (A0 − AA)/A0 × 100, where A0 was absorbance of DPPH and AA was absorbance of the sample.

Antioxidant activity was calculated in relation to standard reference Trolox (Sigma Aldrich, Schnelldorf, Germany) dissolved in methanol (Uvasol^®^ for spectroscopy, Merck, Darmstadt, Germany) to concentration range 0–100 µg/mL. Total antioxidant activity was expressed according to calibration curve as 1 μg of Trolox to 1 mL of the EO sample (TEAC).

### 4.4. Microorganisms

Gram-positive bacteria (*Bacillus subtilis* CCM 1999, *Staphylococcus aureus* subsp. aureus CCM 8223, *Enterococcus faecalis* CCM 4224,), gram-negative bacteria (*Pseudomonas aeruginosa* CCM 3955, *Yersinia enterocolitica* CCM 7204, *Salmonella enterica* subsp. *enterica* ser. Enteritidis CCM 4420, *Serratia marcescens* CCM 8588), and yeasts (*Candida krusei* CCM 8271, *Candida albicans* CCM 8261, *Candida tropicalis* CCM 8223, *Candida glabrata* CCM 8270) were obtained from the Czech collection of microorganisms (Brno, Czech Republic). The biofilm-forming bacterial strain *Pseudomonas fluorescens* was obtained from the fish and *Salmonella enteritidis* was obtained from the sample of meat. Bacteria were identified with 16S rRNA sequencing and MALDI-TOF MS Biotyper. A fungi *P. glabrum*, *P. chrysogenum*, *P. expansum*, and *P. commune* were obtained from grape samples, and were identified by 16S rRNA sequencing and MALDI-TOF MS Biotyper.

### 4.5. Determination of Antimicrobial Activity

Antimicrobial activity of *T. vulgaris* EO was determined by the disc diffusion method. The microbial inoculum was cultivated for 24 h on tryptone soya agar (TSA, Oxoid, Basingstoke, UK) at 37 °C for bacteria and sabouraud dextrose agar (SDA, Oxoid, Basingstoke, UK) at 25 °C for yeasts. The inoculum was adjusted to optical density 0.5 McFarland standard (1.5 × 10^8^ CFU/mL) and 100 μL was added on plates with Mueller Hinton agar (MHA, Oxoid, Basingstoke, UK). Sterile 6 mm discs were saturated with 10 μL of *T. vulgaris* EO and placed on the layer of agar with microbial suspension. Samples were incubated for 24 h at 37 °C for bacteria and 25 °C for yeasts. Two antibiotics (cefoxitin, gentamicin; Oxoid, Basingstoke, UK), and one antifungal (fluconazole; Oxoid, Basingstoke, UK) were used as positive controls for gram-negative, gram-positive bacteria and yeasts, respectively. Disks impregnated with 0.1% DMSO (dimethylsulfoxid, Centralchem, Bratislava, SK) served as the negative control.

An inhibition zone above 15 mm was determined as very strong antimicrobial activity, the inhibition zone above 10 mm was determined as moderate activity, and inhibition zone above 5 mm was determined as weak activity. Antimicrobial activity was measured in triplicate.

### 4.6. Minimum Inhibitory Concentration (MIC)

Microbial inoculum was cultivated for 24 h in Mueller Hinton broth (MHB, Oxoid, Basingstoke, UK) at 37 °C for bacteria and sabouraud dextrose broth (SDB, Oxoid, Basingstoke, UK) at 25 °C for yeasts. A total of 50 μL of inoculum with optical density 0.5 of the McFarland standard was added to a 96-well microtiter plate. Subsequently, the *T. vulgaris* EO was prepared by serial dilution to the concentration range of 400 μL/mL to 0.2 μL/mL in MHB/SDB and 100 μL of suspension was thoroughly mixed with bacterial inoculum in wells. Bacterial samples were incubated for 24 h at 37 °C. Yeast samples were incubated for 24h at 25 °C. MHB/SDB with EO was used as a negative control and MHB/SDB with inoculum was used as positive control of the maximal growth.

For non-adherent microorganisms, the absorbance was measured after the incubation period at 570 nm by Glomax spectrophotometer (Promega Inc., Madison, WI, USA). The MIC of biofilm-forming bacteria was measured with the use of crystal violet. The suspension with non-attached cells was discarded, the wells were washed with distilled water three times, and left to dry at room temperature. A total of 200 μL of 0.1% (*w*/*v*) crystal violet was added to the wells and samples were incubated for 15 min. Subsequently, the wells were repeatedly washed and dried. Stained biofilms were resolubilized with 200 μL of 33% acetic acid [47]. Absorbance was measured at 570 nm. The concentration of EO which absorbance was lower than the absorbance of the maximal growth control was determined as the minimum inhibitory concentration. The test was prepared in triplicate.

### 4.7. Analysis of Differences in Biofilm Development with MALDI-TOF MS Biotyper

The changes of protein spectra during biofilm development after *T. vulgaris* EO addition were evaluated by MALDI-TOF MS Biotyper. The biofilm forming bacteria were added to 50 mL polypropylene tubes with 20 mL of MHB; subsequently, a wooden toothpick and a glass slide were added as a model of different surfaces. The experimental groups were treated with 0.1% *T. vulgaris* EO, and control group samples were left untreated. The samples were incubated at 37 °C on shaker with 170 rpm.

The samples were analyzed after 3, 5, 7, 9, 12, and 14 days. The biofilm samples were taken from a glass slide and wooden toothpick with a sterile cotton swab and were imprinted onto a MALDI-TOF metal target plate. The planktonic cells were obtained from 300 µL of culture medium, cells were centrifuged for 1 min at 12,000 rpm, and the supernatant was discarded. The pellet was three times resuspended in 30 μL of ultrapure water and the suspension was centrifuged for 1 min at 12,000 rpm. In the last step, 1 μL of planktonic cells suspension was applied to a target plate.

The target plate was dried and 1 μL of α-Cyano-4-hydroxycinnamic acid matrix (10 mg/mL) was applied. The samples were processed with MALDI-TOF MicroFlex (Bruker Daltonics) linear and positive mode for the range of m/z 200–2000 after crystallization. The spectra were obtained by an automatic analysis and the same sample similarities were used to generate the standard global spectrum (MSP) and the 19 MSP was generated from the spectra by MALDI Biotyper 3.0 and were grouped into dendrograms using Euclidean distance [40].

### 4.8. Antimicrobial Analysis In Situ on a Food Model

The antifungal effect of the *T. vulgaris* EO vapor phase was evaluated in 0.5 L sterile glass jars (Bormioli Rocco, Italy) on a bread used as a food model. The fungi of *Penicillium* genus were cultivated for 5 days on sabouraud dextrose agar (SDA; Oxoid, Basingstoke, UK) at 25 °C. The cultures were applied to the bread slices (15 × 15 × 1.5 cm) by three stabs. A 6 cm sterile filter paper was placed to the jar lid and 100 µL of *T. vulgaris* EO (62.5, 125, 250, and 500 µL/L diluted in ethyl acetate) were applied. The control group was left untreated. The jars were hermetically sealed and were incubated in the dark for 14 days at 25 °C ± 1 °C.

In situ antibacterial analysis in the vapor phase was tested on *S. marcescens*. Warm MHA was poured into 60 mm Petri dishes (PD) and the lid. Sliced carrots (0.5 mm) were placed on agar. Then, an inoculum was prepared as previously described. *T. vulgaris* EO was diluted twice in ethyl acetate to 500, 250, 125, and 62.5 μL/L and used for sterile filter paper inoculation. The filter paper was placed in for 1 min to evaporate the remaining ethyl acetate, sealed, and incubated at 37 °C for 7 days.

An inhibition of the fungal growth was evaluated by stereological methods. A volume density (Vv) of fungi was estimated using ImageJ software. The stereological grid points of the colonies (P) and substrate (p) were counted. The density of fungal growth was calculated in % according to the formula Vv = P/p × 100. The antifungal activity of EO was expressed as mycelial growth inhibition in % (MGI): MGI = [(C − T)/C] × 100, where C was the density of the fungal growth in the control group and T was the density of the fungal growth in the treatment group [48,49].

In situ bacterial growth was determined using stereological methods. In this concept, the volume density (Vv) of bacterial colonies was firstly estimated using ImageJ software and counting the points of the stereological grid hitting the colonies (P), and those (p) falling to the reference space (growth substrate used). The volume density of bacterial colonies was consequently calculated as follows: Vv (%) = P/p. The antibacterial activity of EO was defined as the percentage of bacterial growth inhibition (BGI) BGI = [(C − T)/C] × 100, where C and T were bacterial growth (expressed as Vv) in the control group and the treatment group, respectively. The negative results represented the growth stimulation.

### 4.9. Statistical Data Evaluation

There was SAS^®^ software version 8 used for data processing. The results of the MIC value (concentration that caused 50% and 90% inhibition in bacterial growth) were determined by logit analysis.

## 5. Conclusions

The main components of *T. vulgaris* essential oil were thymol 48.1%, *p*-cymene 11.7%, 1,8-cineole 6.7%, γ-terpinene 6.1%, and carvacrol 5.5%. The antioxidant activity of the essential oil was 85.2 ± 0.2%, which corresponds to 479.34 ± 1.1 TEAC. We rate this antioxidant activity as high. *T. vulgaris* essential oil had very good antimicrobial effects as well as antibiofilm effects, as observed on various surfaces and detected by MALDI-TOF MS Biotyper. In the test of the antimicrobial activity of the vapor phase of the essential oil, very good antifungal effects against the genus *Penicillium* were observed on the model food (bread). Good effects on inhibition of carrots were also observed in the vapor phase test against *S. marcescens*. There is a possibility of future use of *T. vulgaris* essential oil in extending the shelf life of bakery products and it could find application in the storage of root vegetables. In the case of essential oils with a dominant proportion of volatile components, a stronger effect of the vapor phase is often observed compared to contact application. The vapor phase has a weaker effect on sensory properties than the contact phase. In the future, it would be necessary to evaluate the influence of essential oil in the vapor phase on the sensory properties of model foods.

## Figures and Tables

**Figure 1 plants-10-01959-f001:**
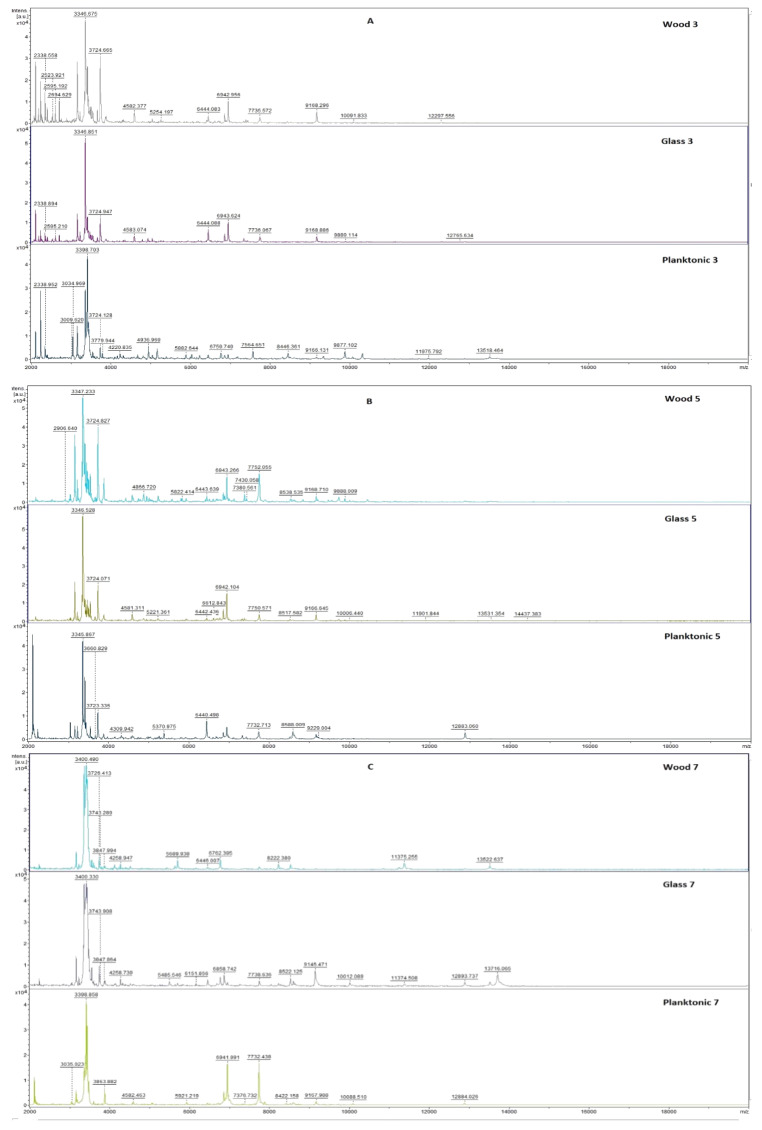
MALDI-TOF mass spectra of *S. enteritidis* biofilm during development after the addition of *T. vulgaris* EO: (**A**) 3rd day, (**B**) 5th day, (**C**) 7th day, (**D**) 9th day, (**E**) 12th day, and (**F**) 14th day.

**Figure 2 plants-10-01959-f002:**
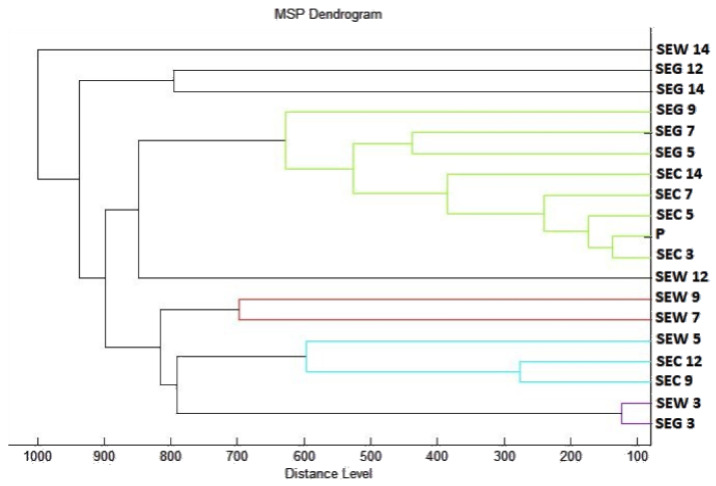
Dendrogram of *S. enteritidis* generated using MSPs of the planktonic cells and the control. SE, *S. enteritidis*; C, control; G, glass; W, wood; and P, planktonic cells.

**Figure 3 plants-10-01959-f003:**
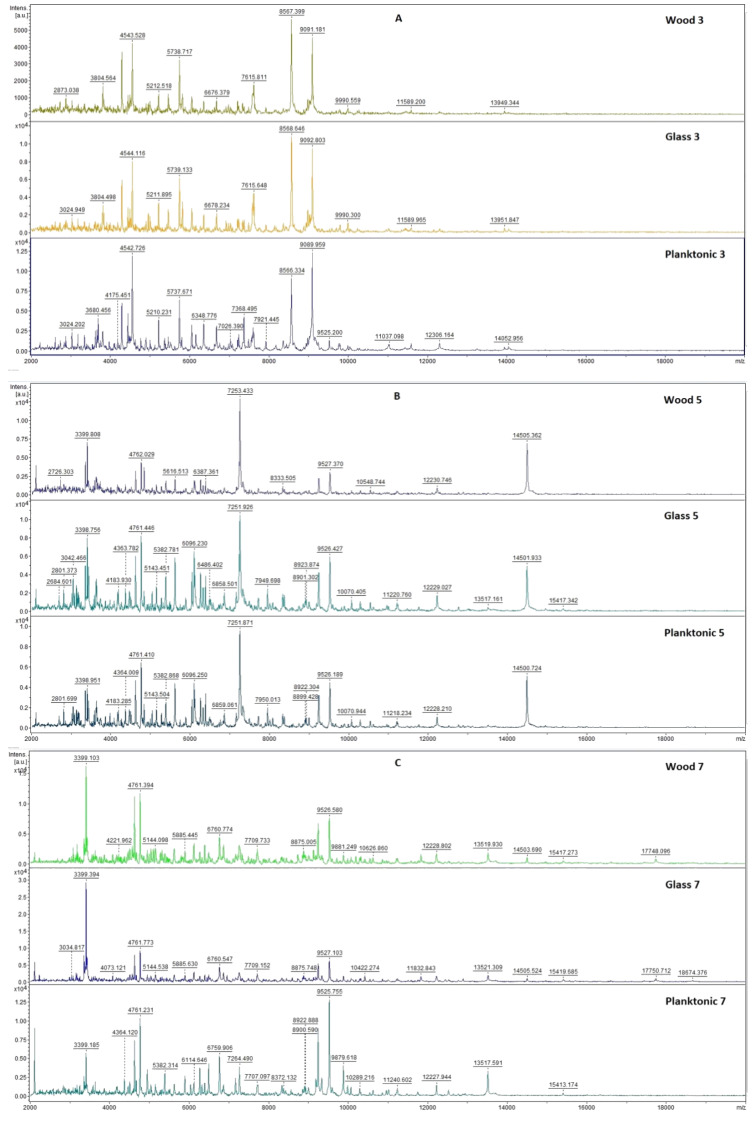
Representative MALDI-TOF mass spectra of *P. fluorescens*: (**A**) 3rd day, (**B**) 5th day, (**C**) 7th day, (**D**) 9th day, (**E**) 12th day, and (**F**) 14th day.

**Figure 4 plants-10-01959-f004:**
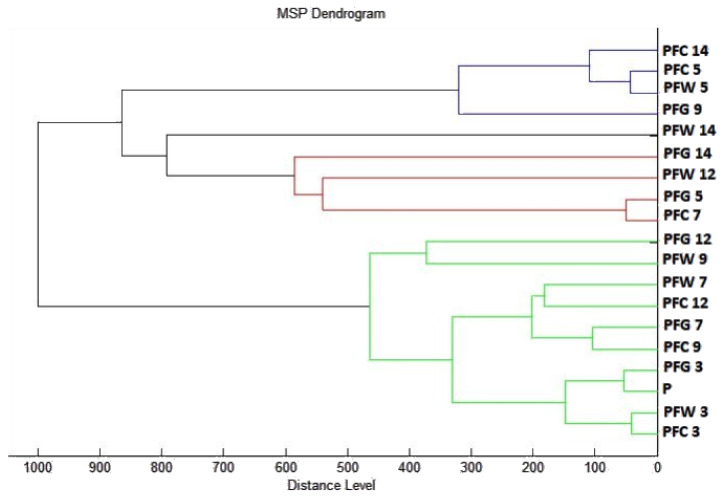
Dendrogram of *P. fluorescens* generated using MSPs of the planktonic cells and the control. PF, *P. fulorescens*; C, control; G, glass; W, wood; and P, planktonic cells.

**Table 1 plants-10-01959-t001:** Chemical composition of essential oil of *T. vulgaris*.

	ERI ^a^	LRI ^b^	Compound ^c^	% ^d^
1	926	930	α-thujene	0.5
2	938	939	α-pinene	2.1
3	948	954	camphene	0.7
4	977	975	sabinene	0.2
5	980	979	β-pinene	0.9
6	992	990	β-myrcene	1.0
7	993	991	3-octanol	tr
8	1004	1002	α-phellandrene	0.1
9	1009	1011	δ-3-carene	tr
10	1016	1017	α-terpinene	0.8
11	1023	1024	*p*-cymene	11.7
12	1028	1029	α-limonene	1.3
13	1033	1031	1,8-cineole	6.7
14	1047	1050	(*E*)-β-ocimene	tr
15	1060	1059	*γ*-terpinene	6.1
16	1088	1088	α-terpinolene	0.3
17	1089	1086	trans-linalool oxide	tr
18	1098	1096	linalool	4.4
19	1148	1146	camphor	1.3
20	1151	1152	menthone	0.2
21	1170	1169	borneol	2.2
22	1178	1177	4-terpinenol	1.9
26	1189	1188	α-terpineol	0.5
27	1245	1244	carvacrol methyl ether	0.3
28	1255	1257	linalool acetate	tr
29	1256	1252	geraniol	tr
30	1286	1285	bornyl acetate	tr
31	1290	1290	thymol	48.1
32	1302	1299	carvacrol	5.5
33	1422	1419	(*E*)-caryophyllene	2.3
34	1507	1505	β-bisabolene	0.4
35	1525	1523	δ-cadinene	0.1
36	1583	1583	caryophyllene oxide	0.3
	total			99.7

^a^ Experimental values of retention indices on HP-5MS column; ^b^ literature values of retention indices; ^c^ identified compounds; ^d^ the percentage of the identified compound; tr = compounds identified in amounts less than 0.1%.

**Table 2 plants-10-01959-t002:** Antimicrobial activity of essential oil of *T. vulgaris*.

Microorganism	Zone Inhibition (mm)	Activity of EO	MIC 50 (µL/mL)	MIC 90 (µL/mL)	ATB
*S. enterica* subsp. *enterica* ser. *enteritidis*	17.00 ± 0.87	***	86.35	121.31	28.00 ± 0.06
*P. aeruginosa*	10.67 ± 0.87	**	103.28	169.19	25.00 ± 0.03
*Y. enterocolitica*	10.56 ± 1.67	**	64.49	71.59	24.00 ± 0.08
*S. aureus*	10.67 ± 1.00	**	16.56	19.26	24.00 ± 0.08
*B. subtilis*	11.33 ± 1.53	**	12.12	16.56	26.00 ± 0.05
*E. faecalis*	10.22 ± 1.30	**	13.85	16.43	25.00 ± 0.08
*C. albicans*	10.56 ± 1.13	**	121.56	159.26	26.00 ± 0.08
*C. krusei*	11.56 ± 1.67	**	165.46	183.21	24.00 ± 0.09
*C. tropicalis*	9.89 ± 1.27	*	135.38	164.43	25.00 ± 0.02
*C. glabrata*	12.22 ± 1.20	**	146.82	169.34	28.00 ± 0.04
*S. marcescens*	17,11 ± 1,27	***	84.27	136.41	22.00 ± 0.04
*S. enteritidis* biofilm	16.67 ± 1.22	***	274.37	311.56	25.00 ± 0.02
*P. fluorescens* biofilm	22.44 ± 1.33	***	97.78	108.82	24.00 ± 0.01

* Weak antimicrobial activity (zone 5–10 mm). ** Moderate inhibitory activity (zone 5–10 mm). *** Very strong inhibitory activity (zone > 15 mm), ATB—antibiotics, positive control (cefoxitin for G^−^, gentamicin for G^+^, fluconazole for yeast).

**Table 3 plants-10-01959-t003:** In situ analysis of the antibacterial activity of the vapor phase of *T. vulgaris* essential oil in bread.

Mycelial Growth Inhibition [%]				
Concentration of EO	62.5 µL/L	125 µL/L	250 µL/L	500 µL/L
Microorganisms				
*P. glabrum*	62.83 ± 10.81 ^a^	80.65 ± 7.22 ^ab^	94.05 ± 7.75 ^b^	99.48 ± 0.74 ^b^
*P. chrysogenum*	22.93 ± 15.53 ^a^	86.07 ± 4.30 ^b^	92.18 ± 8.82 ^b^	98.25 ± 2.48 ^b^
*P. expansum*	75.51 ± 14.95	80.30 ± 3.21	96.38 ± 2.82	100.00 ± 0.00
*P. commune*	82.17 ± 1.37 ^a^	89.87 ± 4.51 ^ab^	93.67 ± 5.58 ^ab^	99.43 ± 0.11 ^b^

Means ± standard deviation. Values followed by different superscript within the same row are significantly different (*p* < 0.05). The statistical differences (*p* < 0.05) between individual EO concentrations were as follows: *P. glabrum*: 62.5 µL/L vs. 250 µL/L and 500 µL/L; *P. chrysogenum:* 62.5 µL/L vs. 125 µL/L, 250 µL/L and 500 µL/L; *P. commune*: 62.5 µL/L vs. 500 µL/L.

**Table 4 plants-10-01959-t004:** Results of in situ analysis of antibacterial activity of the vapor phase of *T. vulgaris* essential oil on carrots.

Bacterial Growth Inhibition [%]				
Concentration of EO	62.5 µL/L	125 µL/L	250 µL/L	500 µL/L
Microorganisms				
*S. marcescens*	57.35 ± 1.43 ^a^	46.78 ± 2.59 ^b^	69.82 ± 3.23 ^c^	87.80 ± 1.41 ^d^

Means ± standard deviation. Values followed by different superscript within the same row are significantly different (*p* < 0.05). The statistical differences (*p* < 0.05) between individual EO concentrations were as follows: 62.5 µL/L vs. 125 µL/L, 250 µL/L and 500 µL/L; 125 µL/L vs. 250 µL/L and 500 µL/L; 250 µL/L vs. 500 µL/L.

## Data Availability

Data is contained within the article.

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
