# Peer review of "Thymus vulgaris Essential Oil and Its Biological Activity"

_plants, 2021, doi:10.3390/plants10091959_

Round 1

Reviewer 1 Report

Galovičová et al., studied the biological activity of Thymus vulgaris Essential Oil. The essential oil composition of T. vulgaris was determined using GC/MS and GC-FID. The antioxidant, antimicrobial, and in situ antifungal activity of this oil were determined. Further, the changes of protein spectra during biofilm development after T. vulgaris EO addition were evaluated by MALDI-TOF MS. The manuscript is of interest to the readers and aims of the journal PLANTS. However, there are some flaws that should be addressed before being accepted for publication.

  1. Line no. 65 – 66, the authors mentioned that “To this date, only a few authors have followed the development and structure of biofilms using the MALDI-TOF MS Biotyper. Include relevant citation
  2. Explain the application of MALDI-TOF MS for studying the development of biofilms in the introduction section.
  3. Include the yield of Thymus vulgaris essential oil in the results section (2.1).
  4. Italicize the scientific names of plants and microorganisms throughout the manuscript.
  5. Compare retention indices with those reported in the literature.
  6. The normal trend for presenting the result of DPPH scavenging activity is the IC50 Because the antioxidant activity of the sample is changed according to its concentration. Better add IC50 value for DPPH activity.
  7. The authors did not use any standard antibiotic to compare the antimicrobial activity of vulgaris essential oil in the Disc diffusion method. The positive or negative control is very important.
  8. Lines 26, 44, 129, 198, 251, 323, 328, 338, 354, 358, 375, 379, and 499 correct the spelling of vulgaris not T. vulgare.
  9. Is there any possibility to present Figures 2 and 3 as 3rd, 5th, 7th, 9th, 12th and 14th day for Wood, Glass Planktonic separately? Then only readers can easily observe the changes in biofilm development with T.  vulgaris oil treatment.
  10. The discussion section is very poor, especially about MALDI-TOF MS. Discuss more about MALDI-TOF MS studies in relation to biofilm development using previously published articles.

Author Response

Galovičová et al., studied the biological activity of Thymus vulgaris Essential Oil. The essential oil composition of T. vulgaris was determined using GC/MS and GC-FID. The antioxidant, antimicrobial, and in situ antifungal activity of this oil were determined. Further, the changes of protein spectra during biofilm development after T. vulgaris EO addition were evaluated by MALDI-TOF MS. The manuscript is of interest to the readers and aims of the journal PLANTS. However, there are some flaws that should be addressed before being accepted for publication.

Point 1: Line no. 65 – 66, the authors mentioned that “To this date, only a few authors have followed the development and structure of biofilms using the MALDI-TOF MS Biotyper. Include relevant citation

 Response: It was added.

 Point 2: Explain the application of MALDI-TOF MS for studying the development of biofilms in the introduction section.

 Response: The explanation was added in L 62-68.

 Point 3: Include the yield of Thymus vulgaris essential oil in the results section (2.1).

Response: Essential oil of Thymus was obtained from producer Hanus, it is commercially produced EO, and the producer did not dispose with the information about the yield.

Point 4: Italicize the scientific names of plants and microorganisms throughout the manuscript.

Response: It was fixed.

Point 5: Compare retention indices with those reported in the literature.

Response: In Table 1 we added literature values of retention indices.

Point 6: The normal trend for presenting the result of DPPH scavenging activity is the IC50 Because the antioxidant activity of the sample is changed according to its concentration. Better add IC50 value for DPPH activity.

Response: We did not compare the different concentrations of the EO, we only used the 100 % EO from the producer, thus we are not able to provide the IC 50 value. For the next analyses we can improve our measurements.

Point 7: The authors did not use any standard antibiotic to compare the antimicrobial activity of vulgaris essential oil in the Disc diffusion method. The positive or negative control is very important.

Response: The positive controls with antibiotics were add to Table 2 and line 345-349.

Point 8: Lines 26, 44, 129, 198, 251, 323, 328, 338, 354, 358, 375, 379, and 499 correct the spelling of vulgaris not T. vulgare.

Response: It was fixed.

Point 9: Is there any possibility to present Figures 2 and 3 as 3rd, 5th, 7th, 9th, 12th and 14th day for Wood, Glass Planktonic separately? Then only readers can easily observe the changes in biofilm development with T.  vulgaris oil treatment.

Response: Group comparison is more suitable for comparing differences between spectra in one day.

Point 10: The discussion section is very poor, especially about MALDI-TOF MS. Discuss more about MALDI-TOF MS studies in relation to biofilm development using previously published articles.

Response: It was added.

Reviewer 2 Report

Plants - 1381299

The authors aimed to determine the composition of the essential oil of Thymus vulgaris and to evaluate its possible biological activities including antioxidant, antimicrobial and inhibition of biofilm.

The introduction is well done and references look appropriate.

Line 40 please use plural (are): there are six chemo…

The composition of the EO is well displayed in table 1.

Results of antioxidant, antimicrobial activity and MIC are reported together in section 2.2.

In section 2.3 they analysed developmental phase of biofilms reporting the results in two very long figures (1 and 3). These results are very detailed and interesting.

Table 2. Please specify the rationale of “weak” “moderate” and “very strong” What are the parameters used? The reader can see the results in the table, do asterisks relate to statistical analysis?

2.5 The authors report data of antimicrobial effect on carrots. In Material and Methods the vapour phase experiments are described using bread slices. Can the authors explain how carrots were used?

3.Discussion

The discussion includes several appropriate references and the authors clearly discuss their results and compare with data available in the literature cited. However, the data present in the literature indicate that most of the results of this paper are not real novelties.

Material and Methods

Lines 269-270 The authors say: “Thymus vulgaris EO of thymol chemotype was obtained from Hanus, s.r.o. (Nitra, Slovakia). The EO was prepared by steam distillation of the partially dried stalk.”

Can the authors supply some information on the chemical composition stated by Nitra s.r.o., or at least the content in thymol stated by the firm? If the supplier never analysed the chemical composition this should be stated.

Please check that all Latin binomials are in Italics.

Author Response

The authors aimed to determine the composition of the essential oil of Thymus vulgaris and to evaluate its possible biological activities including antioxidant, antimicrobial and inhibition of biofilm.

The introduction is well done and references look appropriate.

Point 1: Line 40 please use plural (are): there are six chemo…

Response: It was fixed.

Point 2: The composition of the EO is well displayed in table 1.

Response: Thank you very much for this favorable opinion.

Point 3: Results of antioxidant, antimicrobial activity and MIC are reported together in section 2.2.

In section 2.3 they analysed developmental phase of biofilms reporting the results in two very long figures (1 and 3). These results are very detailed and interesting.

Response: Thank you very much for this favorable opinion.

Point 4: Table 2. Please specify the rationale of “weak” “moderate” and “very strong” What are the parameters used? The reader can see the results in the table, do asterisks relate to statistical analysis?

Response: An explanation is given in section 4.5. It was added to the legend of the table.

Point 5: 2.5 The authors report data of antimicrobial effect on carrots. In Material and Methods the vapour phase experiments are described using bread slices. Can the authors explain how carrots were used?

Response: It was added.

Point 6: Discussion

The discussion includes several appropriate references, and the authors clearly discuss their results and compare with data available in the literature cited. However, the data present in the literature indicate that most of the results of this paper are not real novelties.

Response:

In some parts of the manuscript there is a new idea about the study of T. vulgaris. Some references are used in the discussion, which are not directly related to the observed issues only marginally. In this work, T. vulgaris was used for the first time in such a wider range, and for the first time a biofilm was observed in relation to this essential oil.

Point 7: Material and Methods

Lines 269-270 The authors say: “Thymus vulgaris EO of thymol chemotype was obtained from Hanus, s.r.o. (Nitra, Slovakia). The EO was prepared by steam distillation of the partially dried stalk.”

Response: This is the only information the producer provides, we were trying to enhance their knowledge about the mentioned EO, including the composition ant the other properties.

Point 8: Can the authors supply some information on the chemical composition stated by Nitra s.r.o., or at least the content in thymol stated by the firm? If the supplier never analysed the chemical composition this should be stated.

Response: Hanus s.r.o. lists the main components on its website without a share percentage. So we assume that, they have not the specific information about the composition (only the chemotype was determines).

Point 9: Please check that all Latin binomials are in Italics.

Response: It was fixed.

Reviewer 3 Report

The manuscript is very interesting in particular  the study of biofilm with the MALDI Tof.  Also if  the study the antimicrobial activity of EO against Candida spp is been evaluated using a medium that generally were used for the bacteria and not for the yeasts. Also the temperature of incubation was 25 °C and not  28/30 °C as  is used generally. 

This part must be revised to revalue the results obtained with suitable medium and temperature of incubation for  the yeasts.  

Author Response

The manuscript is very interesting in particular the study of biofilm with the MALDI Tof. 

 Point 1: Also if the study the antimicrobial activity of EO against Candida spp is been evaluated using a medium that generally were used for the bacteria and not for the yeasts. Also the temperature of incubation was 25 °C and not 28/30 °C as  is used generally. 

Response: It was fixed.

Point 2: This part must be revised to revalue the results obtained with suitable medium and temperature of incubation for the yeasts.  

Response: Thank you for valuable comments.

Round 2

Reviewer 1 Report

Galovičová et al., studied the biological activity of Thymus vulgaris Essential Oil. In their study, antioxidant, antimicrobial, antibiofilm activity as well as the chemical composition of the essential oil of Thymus vulgaris were evaluated.

I have read the reply of the authors and the revised manuscript. I can say that the authors answered most of the inquiries about their work. Further, the manuscript has been revised according to reviewers' comments and suggestions. Hence, I recommend this manuscript for publication in the ‘PLANTS’ journal.

Minor comments

  1. Italicize the scientific names of plants and microorganisms in the references section.
  2. Use the full form of botanical name (Thymus vulgaris) when mention in the text for the first time, later use its abbreviated form, T. vulgaris.

Author Response

Galovičová et al., studied the biological activity of Thymus vulgaris Essential Oil. In their study, antioxidant, antimicrobial, antibiofilm activity as well as the chemical composition of the essential oil of Thymus vulgaris were evaluated.

I have read the reply of the authors and the revised manuscript. I can say that the authors answered most of the inquiries about their work. Further, the manuscript has been revised according to reviewers' comments and suggestions. Hence, I recommend this manuscript for publication in the ‘PLANTS’ journal.

Minor comments

  1. Italicize the scientific names of plants and microorganisms in the references section.
  2. Use the full form of botanical name (Thymus vulgaris) when mention in the text for the first time, later use its abbreviated form, T. vulgaris.

The authors are very grateful to the Editor and Reviewer for their valuable comments. We would like to thank the Editor and Reviewer for the time devoted for constructive and important comments to improve our paper. All changes in the manuscript have been done. 

The authors are very grateful to the Reviewer for their valuable comments. We would like to thank the Reviewer for the time devoted for constructive and important comments to improve our paper. All changes in the manuscript have been done. 

Reviewer 3 Report

The manuscript can be publish in this form